# Diagnostic methods for atmospheric inversions of long-lived greenhouse gases

Anna M. Michalak[1], Nina A. Randazzo[1,2], Frédéric Chevallier[3]

[1]Department of Global Ecology, Carnegie Institution for Science, Stanford, California, United States
[2]Department of Earth System Science, Stanford University, Stanford, California, United States
[3]Laboratoire des Sciences du Climat et de l'Environnement, Gif-sur-Yvette, France

*Correspondence to*: Anna M. Michalak (michalak@stanford.edu)

**Abstract.** The ability to predict the trajectory of climate change requires a clear understanding of the emissions and uptake
(a.k.a. surface fluxes) of long-lived greenhouse gases (GHGs).  Furthermore, the development of climate policies is driving a need to constrain the budgets of anthropogenic GHG emissions.  Inverse problems that couple atmospheric observations of GHG concentrations with an atmospheric chemistry and transport model have increasingly been used to gain insights into surface fluxes.  Given the inherent technical challenges associated with their solution, it is imperative that objective approaches exist for the evaluation of such inverse problems. Because direct observation of fluxes at compatible
spatiotemporal scales is rarely possible, diagnostics tools must rely on indirect measures. Here we review diagnostics that have been implemented in recent studies, and discuss their use in informing adjustments to model setup.  We group the diagnostics along a continuum starting with those that are most closely related to the scientific question being targeted, and ending with those most closely tied to the statistical and computational setup of the inversion.  We thus begin with diagnostics based on assessments against independent information (e.g., unused atmospheric observations, large-scale
scientific constraints), followed by statistical diagnostics of inversion results, diagnostics based on sensitivity tests and analyses of robustness (e.g., tests focusing on the chemistry and transport model, the atmospheric observations, or the statistical and computational framework), and close with the use of synthetic data experiments (a.k.a. observing system simulation experiments (OSSEs)).  We find that existing diagnostics provide a crucial toolbox for evaluating and improving flux estimates, but, not surprisingly, cannot overcome the fundamental challenges associated with limited atmospheric
observations or the lack of direct flux measurements at compatible scales.  As atmospheric inversions are increasingly expected to contribute to national reporting of GHG emissions, the need for developing and implementing robust and transparent evaluation approaches will only grow.

## 1 Introduction and the need for diagnostics

The ability to predict the trajectory of climate change requires a clear understanding of the historical and current emissions and uptake (a.k.a. surface fluxes) of long-lived greenhouse gases, and chief among them carbon dioxide ($CO_2$) and methane ($CH_4$), over the Earth's land and ocean regions. For the natural components of the global budgets of these gases,

understanding historical and contemporary flux patterns is needed for elucidating the biogeochemical processes that control flux variability, and therefore the likely evolution of these fluxes under changing climate scenarios (e.g., Friedlingstein et al., 2014). The ability to constrain the anthropogenic components of greenhouse gas budget estimates, on the other hand, is becoming increasingly central to discussions aimed at setting emissions, or emissions reduction, targets at local to global scales (e.g., Pacala et al., 2010).

Direct monitoring of the fluxes of greenhouse gases is only feasible at a limited number of spatial and temporal scales, however. Point sources of anthropogenic emissions can be measured directly at discrete times for example (e.g., Allen et al., 2015; Subramanian et al., 2015; Zimmerle et al., 2015), while biospheric fluxes over land can be continuously monitored at plot scale (i.e. from a few hectares to a few $km^2$, depending on sensor height) using the eddy covariance technique (e.g., Baldocchi et al., 2001; Law et al., 2002), and ocean fluxes can also be deduced locally from the difference between the

partial pressure of $CO_2$ measured in seawater and that in the overlying air (e.g., Takahashi et al., 1993, 2002). At the global scale, a network of observation sites tracks the global growth rate of atmospheric concentrations of greenhouse gases, and gives broad insight into the temporal (e.g., seasonal, interannual) and spatial (e.g., hemispheric, latitudinal) signatures of net greenhouse gas emissions (e.g., Tans et al., 1990; Steele et al., 1992).

The target applications listed in the first paragraph, however, require an understanding of fluxes at intermediate scales, e.g.,

from urban to biome to national to continental. Direct observations of fluxes are not feasible at these scales, and gaining an understanding of flux budgets and controlling processes at these scales therefore invariably depends on a process of either "upscaling" small-scale flux observations, or "downscaling" large-scale information provided by atmospheric concentration measurements. Upscaling strategies range from the implementation of mechanistic models calibrated using plot-scale flux observations (e.g., Richardson et al., 2012; Schaefer et al., 2012), to the development of statistical or machine learning

approaches for elucidating dominant patterns (e.g., Beer et al., 2010; Jung et al., 2011), to the combination of fine-scale flux measurements with activity data (e.g., fuel consumption for anthropogenic emissions, or burnt area for fire emissions) as the basis of emissions inventories (e.g., van der Werf et al., 2006; Jeong et al., 2014; Lyon et al., 2015). Downscaling strategies, on the other hand, most typically involve the solution of an inverse problem to elucidate spatially and temporally resolved flux information from upwind and downwind observations of atmospheric greenhouse gas abundance (e.g., Enting et al.,

30 2002).

Inverse problems that couple atmospheric observations of greenhouse gas concentrations with an atmospheric chemistry and transport model in order to gain insights into underlying flux patterns have been used since the late 1980s (e.g., Enting and Mansbridge, 1989; 1991). While the observational network has expanded and the statistical and numerical methods have

become more sophisticated (e.g., Ciais et al., 2010a; Michalak, 2013; Miller and Michalak, 2017; Houweling et al., 2017), the underlying principles have remained largely unchanged. Spatiotemporal flux patterns at the Earth's surface lead to spatial and temporal gradients in atmospheric concentrations of greenhouse gases. The inverse problem then amounts to using those gradients to recover information about the flux patterns. From a scientific perspective, an additional goal is often

to also gain insight into the enviro-climatic factors driving these patterns (e.g., Gourdji et al., 2012; Fang and Michalak, 2015; Miller et al. 2014, 2016b). Although the principle is simple, the atmospheric inverse problem is ill-conditioned because the diffusive nature of atmospheric transport means that relatively small variations or errors in observed or modelled atmospheric concentrations can correspond to relatively large differences or errors in the inferred flux quantities and patterns. In addition, the atmospheric inverse problem is often under-determined because the sparse observational coverage

precludes the possibility of resolving fluxes (spatially and temporally) at all the scales that are of scientific or policy interest, as well as at all the scales to which atmospheric observations are locally sensitive.

Given the high scientific and policy value of accurate greenhouse gas budgets, the growing role of atmospheric inverse problems to obtain these budgets at relevant scales, and the inherent technical challenges associated with the solution of these inverse problems, it is imperative that objective approaches exist for evaluating the scientific value and accuracy of

inverse modelling estimates of greenhouse gas fluxes. Here, we review diagnostics that have been implemented in recent studies, and discuss their use in informing adjustments to model setup. We have structured the review in a manner that we hope will be useful to novices and specialists alike. We present a relatively comprehensive survey of recent approaches, in order to provide a detailed representation of the state-of-the art for specialists. At the same time, we have organized the review around high-level categories in order to help guide researchers who are newer to the field and provide an entry point

for further inquiry via the cited studies.

Fundamentally, the emphasis of diagnostic tools should be on the scientific value of insights that are based on the solution of an atmospheric inverse problem. This quality control approach (i.e., the evaluation of the flux estimates) also has to be complemented by quality assurance (i.e., the evaluation of the estimation process that yielded the flux estimates). Indeed, the solution of atmospheric inverse problems invariably involves a series of decision points including, but not limited to, (1) the

choice of the atmospheric observations to be used, (2) the choice of the atmospheric chemistry and transport model to be implemented, (3) the choice of a statistical framework for defining an objective function that captures the relative contribution of atmospheric observations, the chemistry and transport model, and any prior information in informing flux patterns, and (4) the choice of a numerical framework for the solution of the inverse problem. Each of these choices will have a direct impact on estimates. It is therefore also imperative to have diagnostic tools that can evaluate the self-

consistency of the modelling and statistical assumptions specific to the choices made in the setup of the inverse problem. In other words, at a minimum, the ultimate estimates must be consistent with the assumptions inherent to the specific modelling setup that was implemented.

## 2 Challenges of diagnosing atmospheric inversions

Having established the need for diagnostic tools to assess atmospheric inverse modelling results, the question then becomes one of identifying appropriate diagnostics, metrics, or benchmarks. As discussed in the last section, however, direct observation of greenhouse gas fluxes is not possible at the space and time scales targeted by atmospheric inversions. This is in part because inversion systems for long-lived greenhouse gases are run over time periods ranging from weeks to decades to capture the long dispersion times of tracers in the atmosphere and to capture temporal variability in fluxes. These long timespans are achieved at the expense of relatively coarse horizontal resolutions, ranging from tens of kilometres to one or more degrees, such that the large gap between flux measurements and inverse model scales precludes direct evaluation of inverse modelling results. This gap is filled only rarely by some regional inversions (e.g., Lauvaux et al., 2009, Meesters et al., 2012). This means that there is a basic lack of independent measures of flux to assess inverse modelling estimates.

Diagnostic tools used for assessing inverse modelling estimates must therefore rely on other indirect measures or information about the fluxes to be estimated. Such measures and information should, in principle, be independent from the information used in the solution of the original inverse problem. A natural choice might then be to use additional atmospheric concentration data not assimilated in the original inverse problem, because, as noted earlier, gradients in atmospheric greenhouse gas concentrations are themselves the result of underlying flux patterns. Given the ill conditioned and typically under-determined nature of the atmospheric inverse problem, however, it is often desirable to use as much information (i.e. data) as possible to inform the initial solution of the inverse problem, in order to gain the deepest and most precise insights possible about flux patterns. This goal, however, is at odds with the desire to keep some independent flux-relevant observations for diagnosing the estimates obtained from the inversion. Although this problem is not unique to the solution of atmospheric inverse problems, it is certainly particularly salient in this context. Two examples follow.

In some ways, numerical weather forecasting (e.g., Kalnay et al., 2003) bears some resemblance to the flux estimation problem, as they both rely on atmospheric observations and a numerical representation of atmospheric dynamics. In both cases, the ability to diagnose the accuracy and precision of estimates is of high value. Key differences emerge upon closer examination, however. First, the target quantities predicted/estimated in numerical weather prediction, such as temperature, precipitation, and barometric pressure, are ones that can also be measured directly at a large number of locations, via both *in situ* and remote sensing observations, making a comparison to direct benchmarks feasible (e.g., ECMWF, 2016). Although it is technically true that in some cases a scale mismatch still occurs (e.g., a thermometer cannot measure the "average" temperature over a computational grid box), the quantities of interest are less likely to display the strong multi-scale heterogeneity that makes eddy covariance flux observations ill-suited for diagnosing grid-scale inverse-model-derived flux estimates at much coarser spatial resolution. Second, whereas atmospheric inverse problems aim to infer/estimate historical flux distributions that were never observed directly, the accuracy and precision of numerical weather forecast estimates can largely be verified, evaluated, and diagnosed simply by waiting for weather patterns to unfold. This is perhaps best

illustrated through the long-standing comparisons of forecast skill among the world's weather forecasting bureaus (Simmons and Hollingsworth, 2002; WMO-LCDNV, 2016).

Another useful example is that of the development of retrieval algorithms for remote sensing observations of atmospheric constituents (e.g., Rodgers, 2000). Let us take as a prototypical example the process of obtaining estimates of column-integrated dry air mole fractions of atmospheric carbon dioxide ($X_{CO2}$) from the spectrum of reflected sunlight measured by the Orbiting Carbon Observatory (OCO-2) space-borne instrument (e.g., Crisp et al., 2012). In this case, the observations are radiances at specific wavelengths within the spectrum of reflected light, with a focus on specific absorption bands that are observed at high spectral resolution. The forward problem involves the solution of radiative transfer equations. The target variable of primary interest is $X_{CO2}$. This problem has analogies to the flux estimation problem in that the column-integrated $CO_2$ concentrations cannot be measured directly per se. A key difference, however, is that a number of validation datasets are available to help diagnose the retrieval algorithm (e.g., Osterman, 2011). These include, among others, observations from ground-based remote sensing instruments (that look up at the sun, rather than down at the Earth, e.g., Wunch et al. (2011)), and targeted campaigns of *in situ* airborne observations that can capture $CO_2$ concentration variability within a portion of the atmospheric column (e.g., Tadić et al., 2014; Frankenberg et al., 2016). Unlike in the flux estimation problem, there is no direct conflict between using these additional measurements for validation / diagnosis versus using them to directly inform the solution of the inverse problem itself, as there is no clear mechanism by which these additional observations could be routinely incorporated within the core retrieval algorithm, although they can be used for additional empirical bias correction.

Overall, then, while the need for diagnostics to evaluate the scientific validity and statistical self-consistency of flux estimates derived via the solution of atmospheric inverse problems is clear, this need poses very substantial challenges. These include the lack of independent measures of flux at comparable spatiotemporal scales, and the inherent dilemma between using available atmospheric observations for estimation versus validation. These features make the process of developing and implementing diagnostics particularly challenging, and fundamentally different from the challenges observed in other fields that might at first glance appear to be somewhat analogous.

## 3 Overview of existing diagnostics

Researchers have taken a number of approaches in tackling the challenges associated with the development of diagnostics that are both practical, given the unavoidable limitations in available data, and genuinely informative, in terms of assessing the accuracy and precision of flux estimates. Here we describe existing diagnostics that have been used as part of inverse modelling efforts. We focus primarily on diagnostics that evaluate the validity and self-consistency of the inversion setup, rather than on diagnostics designed to assess the information content of specific data sets. We also discuss how diagnostics are used to inform adjustments to model setup and the trade-offs inherent to alternative possible approaches to model evaluation. We focus primarily on examples from papers published between 2010 and 2016, and on papers that present

recent applications of specific diagnostics rather than on the studies where these diagnostics were originally introduced. We do so in order to get a contemporary snapshot of approaches that are currently being used for diagnosing atmospheric inversions. The groupings of diagnostics are ordered here by starting with diagnostics that are most closely related to the actual scientific problem or question being targeted by the inversion, to those that are most closely tied to the statistical and computational setup of the inversion framework itself. More fundamental overriding questions about the types of insights that the range of currently available diagnostics can (or cannot) actually provide are then discussed in Section 4.

## 3.1 Assessment against independent information

The most natural starting point for assessing the solution of an atmospheric inverse problem is through evaluation against independent information. Although, as discussed in earlier sections, direct observations of surface fluxes are seldom available at compatible scales, at least two additional avenues are available. The first is to evaluate flux estimates against unused atmospheric observations, whether from in situ monitoring or remote sensing. This is accomplished through the solution of the "forward" problem, which translates estimated fluxes into modelled atmospheric concentration fluctuations. The second is to compare estimates against any available large-scale scientific constraints. This approach can be challenging especially when large-scale constraints are themselves uncertain.

### 3.1.1 Evaluation against unused atmospheric observations

If any atmospheric observations are available that have not been used as a constraint in the solution of the inverse problem, they can be leveraged to evaluate final flux estimates. To do so, final flux estimates are used as an input into the atmospheric chemistry and transport model used as part of the inversion, and predicted concentrations at the times and locations of the additional available atmospheric observations are then compared to the measured concentrations. These additional observations can be of several types, and inform the inversion setup in various ways, given differences in vertical information, spatial coverage, and precision.

Evaluating inversion results constrained by *in situ* observations using independent surface or satellite total column measurements can provide additional information about regional fluxes. The much broader spatial coverage of satellite observations makes it possible to assess flux estimates at large spatial scales, and thus can help to identify large-scale spatial biases that are related to a lack of *in situ* coverage in some regions (e.g., biases in the latitudinal gradient or over land versus ocean) (e.g., Lindqvist et al., 2015). However, it is important to note in the context of these comparisons that the satellite retrievals themselves may have regional biases, as will be discussed later.

Conversely, for inversions constrained by satellite observation of total column concentrations, evaluating results using *in situ* measurements can reveal errors in the column-constrained system's ability to reproduce surface fluxes, which can be related to aspects of the retrieval (such as biases) or to the transport model's representation of boundary layer dynamics (e.g., Locatelli et al., 2015; Cressot et al., 2014).

Comparisons to independent measurements can also be used to isolate transport errors from the other confounding errors. For example, comparing the total column mixing ratios simulated based on posterior flux estimates obtained using surface data to independent observations of total column mixing ratios can diagnose a transport model's skill in simulating the seasonality of the tropopause height and of the stratospheric partial column (e.g., Houweling et al., 2014). Performing this type of assessment for multiple inversions constrained by different types of measurements but using the same transport model can provide insight into whether seasonal biases in the inversion are caused by seasonal biases in an observing system or to seasonal biases in the transport model (e.g., Houweling et al., 2014). More generally, vertical transport bias can be assessed by comparing the vertical gradients of posterior vertical profiles to those of observed profiles (e.g., Pickett-Heaps et al., 2011; Saeki et al., 2013b; Liu and Bowman, 2016), because vertical gradients provide information about vertical mixing and convection.

More broadly, evaluation against all types of independent atmospheric observations provides an additional window into the degree to which estimated fluxes capture key features of the atmospheric signal, such as the seasonal cycle, latitudinal gradients, or regional patterns of concentrations (e.g., Zhang et al., 2014; Jiang et al., 2014; Diaz Isaac et al., 2014; Pandey et al., 2016; Liu and Bowman, 2016; Johnson et al., 2016).

### 3.1.2 Evaluation at aggregated scales against large-scale scientific constraints

The accuracy of inversion-derived flux estimates and the validity of the overall inversion framework can be assessed, at large scales, based on existing understanding of carbon cycle and atmospheric dynamics. This type of evaluation may involve comparisons of the inversion-derived estimates to existing information about flux magnitudes at large scales, about the overall direction of the net flux in a region (i.e. emission vs. uptake), or about flux seasonality. Care must be taken, however, for the approach not to become circular, i.e. for inversion results not to be evaluated by comparing them to assumed features of the very processes that the inversion is trying to inform.

In the simplest case, spatially aggregated posterior fluxes can be assessed based on expert knowledge of the system. For example, methane emissions in regions dominated by natural gas extraction, urbanization, wetlands, or cattle feedlots are expected to substantially outweigh soil methane uptake, and negative estimated emissions in such regions would point to errors in the inversion (e.g., Berchet et al., 2013). Similarly, global decadal atmospheric growth rates and latitudinal gradients of greenhouse gases are well constrained by long-term baseline observations (e.g., Conway et al., 1994), and posterior flux estimates can be evaluated against such large-scale constraints (e.g., Cressot et al., 2014). Evaluation against observed latitudinal gradients provides information not only about global total fluxes, but can also inform the accuracy of the representation of inter-hemispheric transport, although more so for gases with limited uptake at the Earth surface (e.g., Thompson et al., 2014). This comparison is especially helpful when performed using both surface and upper-troposphere or total column concentrations, because this makes it possible to assess how both meridional and vertical transport are represented (e.g., Thompson et al., 2014).

More broadly, inversion-derived fluxes can be compared against independent estimates of fluxes for comparable regions, although the fact that both the inversion-derived and the independent estimates of fluxes are uncertain must be recognized. For example, the fraction of the global $CO_2$ sink attributable to land versus ocean can be compared between inversions and independent model or mass-balance estimates (e.g., Le Quéré et al., 2015). For specific regions and periods, inversion results can also be compared against detailed inventory estimates of fluxes (e.g., Lauvaux et al., 2012; Schuh et al., 2013). A third example (noted already in Section 3.1.1) is the comparison of large-scale seasonal cycles of modelled trace gas concentrations to observations. For inversions constrained by remotely sensed data, checking for consistency in seasonal cycles between observations, estimates from a satellite-data-constrained-inversion, and estimates from an *in-situ*-data-constrained inversion may draw attention to the need for seasonal bias correction in the observations, while also exploring other potential causes of regional or seasonal bias, such as seasonal biases in vertical transport (e.g., Houweling et al., 2014). Lastly, bottom-up studies also provide regional budget estimates at the annual or pluriannual scale that can be compared to inverse modelling results (e.g., Gourdji et al., 2012; Miller et al., 2013, 2014). The comparison may reveal convergence (e.g., Ciais et al., 2010b) or divergence (e.g., Chevallier et al., 2014; Miller et al., 2013, 2014) of the estimates. However, the attribution of any divergence remains subjective, given the uncertainty of the bottom-up estimates themselves (e.g., Chevallier et al., 2014; Reuter et al., 2014; Gourdji et al., 2012).

Finally, large dipoles in estimated fluxes between large regions can point to a lack of observational constraint for certain regions, to overfitting of the observations that do exist, and/or to biases in large-scale transport (e.g., Alexe et al., 2015; Nassar et al., 2011). The presence of flux dipoles can, however, also be representative of real spatial flux patterns, and sensitivity tests focusing on factors such as the coverage of observational constraints can help to evaluate such patterns in posterior fluxes (e.g., Cressot et al., 2014; Rivier et al., 2010) (see also Section 3.3).

## 3.2 Statistical diagnostics of inversion results

Rather than comparing flux estimates against independent information directly, a second set of strategies focuses instead on assessing whether the prior and posterior flux estimates, uncertainties, and covariances are consistent with the assumptions built into the design of the implemented inversion framework. These strategies thereby focus on statistical self-consistency of the inversion setup, and in this way can point to discrepancies that can signal unreliable results.

The majority of inverse modelling approaches used for greenhouse gas flux estimation leverage a combination of prior information and an observational constraint. Within the mathematical framework of the inversion, the uncertainty and spatiotemporal covariance structure of the prior information (i.e., prior error statistics), as well as the reliability with which the researchers expect to be able to reproduce the atmospheric observations (i.e., model-data-mismatch statistics), are represented through error covariances. These error covariances, the prior information, the observational data, and the chemistry and transport model are then also used to quantify the uncertainty associated with posterior estimates (see e.g., Rayner et al., (2016) for a detailed discussion). This framework provides an opportunity to evaluate the statistical self-consistency of the inversion setup.

For example, under the assumption of Gaussian and unbiased errors and for a given set of assumptions about error correlations, the sum of squared errors follows a chi-squared distribution with a known number of degrees of freedom; for this reason, posterior errors can be used to evaluate or scale assumed prior error variances (e.g., Michalak et al. 2005; Desroziers, 2006; Wu et al., 2013; Lauvaux et al., 2016; Cressot et al., 2014). In some cases, deviations between

concentrations modelled based on posterior fluxes and atmospheric observations not included in the original inversion can be used for this purpose (e.g., Chevallier and O'Dell 2013). This approach can also be used to assess how model-data mismatch errors vary seasonally (e.g., Gourdji et al., 2012; Kim et al., 2011). Also, the very high resolution of some regional inversions and the availability of plot-scale flux measurements make it possible to validate the posterior uncertainty of fluxes directly in some cases (e.g., Broquet et al., 2013).

The spatial and temporal autocorrelation of posterior errors can also be used to inform model setup (Diaz Isaac et al., 2014) or to assess the identifiability of underlying fluxes (Yadav et al., 2016).

Other than assessing self-consistency, statistical diagnostics can also be used to quantify the error reduction (or information gain) made possible by the assimilation of atmospheric observations. In this approach, posterior uncertainties are compared to prior uncertainties. In cases where the explicit quantification of posterior flux uncertainties is prohibitively

computationally expensive, it can also be approximated through approaches such as the use of a Monte Carlo ensemble of inversions in which model parameters are perturbed for each run (e.g., Chevallier et al., 2007; Cressot et al., 2014; Pandey et al., 2016). More simply, the deviations between atmospheric observations not included in the inversion and modelled concentrations based on posterior vs. prior fluxes can be used as a measure of error reduction (e.g., Liu and Bowman, 2016; Johnson et al., 2016; Lauvaux et al., 2016).

**3.3 Sensitivity tests and analysis of robustness**

The validity and robustness of inversion-derived estimates can also be assessed through sensitivity tests. These tests involve running additional inversions where one or several components have been altered. The most common of these are changes to the chemistry and transport model used to translate fluxes into atmospheric concentrations, changes to the set of atmospheric observations used to constrain flux estimates, and changes to the implemented statistical or computational framework.

Examples of the latter include changes to prior estimates, boundary conditions, and flux spatiotemporal resolutions. Results shed light on the degree to which results are robust to specific implementation choices.

**3.3.1 Chemistry and transport model**

Recently, as inversions have become more sophisticated, transport model sensitivity tests have become more computationally expensive. As a result, it has become more difficult to assess the impact of model choice on inversion

results (e.g., Gurney et al., 2002; Baker et al., 2006). Applications focusing exclusively on synthetic data are covered in Section 3.4, while here we present a few examples that included real observations.

Examining the effect of the choice of a chemistry and transport model can lead to various insights. For example, the transport model used by an inversion may be run using different boundary layer schemes to assess how the representation of vertical mixing affects the interpretation of assimilated data (e.g., Peters et al., 2010). Another aspect is the impact of the spatial resolution of the transport model, and particularly the use of finer grids within mesoscale domains versus the coarser grids typical of global transport models. For example, including a finer-scale nested grid and changing the transport representation at these finer scales provides information about the effect of transport representation at scales finer than the grid scale of global transport models (e.g., Rivier et al., 2010). In addition, posterior meridional concentration gradients can be compared across inversions that use different global transport models to assess the effect of interhemispheric transport (e.g., Thompson et al., 2014).

The implementation of more than one transport model in a forward run can also shed light on consistent differences in the ability to represent observed atmospheric concentration signals, seasonal cycles of mixing ratios, or vertical profiles (e.g., Pillai et al., 2012; Diaz Isaac et al., 2014).

### 3.3.2 Atmospheric observations

Performing inversion sensitivity tests in which only the constraining observational data set is changed between inversions can shed light on the impact of various observations on flux estimates, and therefore on their relative information content with regard to underlying fluxes, and also makes it possible to assess the extent to which conclusions are robust to the choice of observations used to constrain the inversion.

For example, a major effort has been made to quantify the effects of including remotely sensed observations (specifically, satellite retrievals) as an additional constraint beyond *in situ* observations. This is distinct from the applications discussed in Section 3.1.1, where remote sensing observations were not included in the inversions, but were instead used to evaluate inversion-derived flux estimates. Satellite data provide the benefit of broader spatial coverage than *in situ* measurements, potentially informing fluxes in regions not well constrained by current *in situ* networks. However, the informational value and robustness of the information provided by satellite observations is still the subject of ongoing research, and thus their use as constraints in inversions requires special consideration of the impacts of any potential biases. Several studies have included satellite total column or mixing ratio data as an additional constraint on a model otherwise constrained only by *in situ* concentration measurements, to determine whether remotely sensed total column concentrations provide a significant amount of additional information (e.g., Alexe et al., 2015; Houweling et al., 2014; Nassar et al., 2011; Pandey et al., 2016; Saeki et al., 2013a). An inversion constrained only by *in situ* measurements may also be compared to an inversion constrained only by satellite measurements (e.g., Cressot et al., 2014). The spatial distribution and magnitude of fluxes and the source/sink status of particular regions are often the major posterior features compared between inversions constrained by different subsets of available data (e.g., Alexe et al., 2015; Cressot et al., 2014; Houweling et al., 2014; Nassar et al., 2011). The differences in the geographical flux patterns can be attributed through the use of various methods focusing on quantifying the information content and geographical coverage of satellite data. The relative information content of the

different observational datasets can be quantified via the degrees of freedom (a metric based on posterior error covariances) provided to the inversion (see e.g., Rodgers 2000), whereby data sets that represent a stronger constraint provide more degrees of freedom (e.g., Nassar et al., 2011). The constraint provided for specific regions by observations with extensive geographical coverage can also be qualitatively analysed by creating visualizations of the sensitivity to fluxes from a certain region (e.g., Nassar et al., 2011). If satellite retrievals provide a large increase in coverage over a particular region, then this method may help to explain large changes in posterior fluxes in upwind areas.

In addition, the robustness of conclusions about flux distributions derived from satellite observations can be explored by using alternative sets of satellite-derived observations. Studies have checked for agreement in posterior fluxes for inversions run using different satellite instruments and retrieval algorithms (e.g., Alexe et al., 2015; Chevallier et al., 2014; Takagi et al., 2014). The effect of the bias correction scheme used for satellite retrieval post-processing has also been a subject of several sensitivity studies (e.g., Houweling et al., 2014; Alexe et al., 2015; Nassar et al., 2011; Cressot et al., 2014, Basu et al., 2013).

Sensitivity tests based on inversions constrained by different subsets of available observations have been used to examine the incremental gain in information obtained by expanding the *in situ* observation network. Such experiments can be used to estimate the uncertainty reduction (see Section 3.2) that could potentially be achieved by assimilating more observations over or downwind from poorly constrained regions, as well as the effects of a more extensive observational network on the estimated spatial and temporal variability of fluxes (e.g., Butler et al., 2010; Saeki et al., 2013b; Kadygrov et al., 2015; Jiang et al., 2014; Peters et al., 2010). They can also be used to determine the value of episodic versus continuous observations (e.g., Peters et al., 2010). These sensitivity tests can also determine whether regions with strong fluxes, such as the "dipoles" discussed in Section 3.1.2, are simply due to a relative lack of constraint for certain regions (e.g., Rivier et al., 2010).

Last, sensitivity tests have also been used to examine the potential role of bias of *in situ* measurements at specific site. In such studies, an offset is added to specific observations, and the results of the control inversion and the inversion with the offset can be compared to determine the effect of potential biases on the posterior flux field (e.g., Peters et al., 2010; Masarie et al., 2011).

### 3.3.3 Statistical and computational framework

Sensitivity tests can be used to explore the impact of the statistical assumptions and computational framework used in inversions.

For example, the impact of assumptions about the statistical representation of prior errors and model-data mismatch errors can be examined by performing multiple inversions, as can the impact of approaches aimed at optimizing these error statistics (e.g., Bousquet et al., 2011; Cressot et al., 2014; Wu et al., 2013; Ganesan et al., 2014; Berchet et al., 2013). Sensitivity tests may also be run on other statistical parameters such as the assumed correlation length of fluxes (Corazza et al., 2011).

Another key aspect of regional inversions that can be explored through sensitivity tests is the impact of the choice of a dataset used to represent background concentrations of greenhouse gases entering the model domain. This can be done through the implementation of alternative boundary conditions, and/or the exploration of the impact of uncertainty in individual sets of boundary conditions (e.g., Göckede et al., 2010b; Bréon et al., 2015; Schuh et al., 2010; Gourdji et al.,

5    2012).

Similar to the case of boundary conditions, inversions aiming to isolate one component of greenhouse gas budgets (e.g., biospheric $CO_2$ in the case of $CO_2$ inversions) must rely on pre-existing estimates of other components of the budget (e.g., fossil fuel $CO_2$ emissions). The impact of the choice of an estimate can be explored through sensitivity tests (e.g., Peylin et al., 2011; Peters et al., 2010).

The choice of a model or data set to be used as an *a priori* estimate in Bayesian inversions is another source of uncertainty in the inferred fluxes, particularly in areas where the observation constraint is weak. Inversions using alternative inventories or process-based models with different spatial and seasonal flux patterns as priors can be compared in terms of the spatial and temporal distributions of the posterior fluxes to assess the robustness of flux estimates (e.g., Kim et al., 2011; Göckede et al., 2010b; Bergamaschi et al., 2015; Corazza et al., 2011; Peters et al., 2010).

A final example is the use of sensitivity tests to explore the effect of the spatial and temporal aggregation and resolution of the unknown fluxes in the modelling framework. The impact of the choice of flux regions, model grid resolution, model grid nesting, or model time step can all be explored (e.g., Rivier et al., 2010; Göckede et al., 2010a; Kim et al., 2014; Peters et al., 2010).

### 3.4 Synthetic data experiments

Observing system simulation experiments (OSSEs) are studies in which synthetic observations are constructed at observation times and locations using a prescribed set of fluxes and a chemistry and transport model. These synthetic observations are then used instead of actual observations as data constraints on an inversion. OSSEs are particularly useful for diagnostics because the "true" transport and fluxes are known and can be manipulated. These types of studies constitute a necessary but certainly not sufficient condition for ensuring a good inversion setup, as many complexities of inversions using real

observations can only be approximated within a synthetic data experiment context. OSSEs have become a key component of inversion model development, especially as models have become more complex.

Because the "true" fluxes are known in an OSSE, various metrics can be used to assess how well the inversion can recover fluxes. OSSEs can be used to quantify the magnitude and geographical distribution of uncertainty that stems from specific errors or assumptions in the inversion framework, such as transport model errors (e.g., Houweling et al., 2010; Berchet et al.,

2015), spatiotemporal flux patterns within regions (e.g., Berchet et al. 2015), biased priors (e.g., Berchet et al. 2015), flux spatiotemporal resolutions (e.g., Wu et al., 2011), or parameter choices within computational data assimilation systems (e.g., Miyazaki et al., 2011, Chatterjee et al. 2012). Posterior flux errors and error covariances can be used to assess the impact of modelling simplifications or data limitations on the accuracy and precision of flux estimation (e.g., Berchet et al., 2015;

Gourdji et al., 2010). OSSEs can also be used to understand sources of bias through a simple differencing of posterior and "true" fluxes (e.g., Locatelli et al., 2013; Thompson et al., 2011; Basu et al., 2016; Bloom et al., 2016). Similar tests can be run to determine the effects of observational biases and mistuning of error statistics on the accuracy of posterior estimates (e.g., Baker et al., 2010).

OSSEs can also be used to determine the sensitivity of inversions to transport errors. The model-data mismatch may be compared between an inversion that uses the "true" transport to calculate the sensitivity matrix versus that of an inversion that uses a different transport model (e.g., Chevallier et al., 2010; Houweling et al., 2010; Berchet et al., 2015; Locatelli et al., 2013). Assuming that the difference in performance between these two transport models is comparable to the difference between transport models used in real-data inversions, the inversion with inconsistent transport can be compared to the

inversion with consistent transport to determine how much the inconsistencies in transport affect the inversion. A similar test can be conducted simply by adding transport or chemistry errors to the pseudo-observations for one run of the model (e.g., Gourdji et al., 2010; Baker et al., 2010; Thompson et al., 2011). In addition, the meteorological forcing field may be perturbed independently of the transport model itself, to determine how the underlying meteorological assumptions affect the inversion; this is particularly important because the meteorology is often not optimized for transport runs (as noted by

Berchet et al., 2015).

OSSEs are also useful for determining the sensitivity of the inversion to the choice of priors. Within a Bayesian inversion, perturbations of prior fluxes from the "true" fluxes in terms of spatial distribution, temporal distribution, and flux magnitude by region can be used for a synthetic data sensitivity test (e.g., Berchet et al., 2015). This type of study is useful for determining prior-related biases in cases when the bottom-up inventories for a particular trace gas in the model domain are

highly uncertain.

OSSEs can also provide information about how much information can be obtained from the current observational network. Pseudo-observation sites and types of data (for example, mixing ratios, profiles, column averages, or isotopic signatures from flask samples) can be added or taken away from the inversion to determine how the density and distribution of observations affect the precision and accuracy of the posterior flux field (Villani et al., 2010; Miyazaki et al., 2011;

Hungershoefer et al., 2010; Shiga et al., 2013; Basu et al., 2016; Bloom et al., 2016). In addition, the ability of existing monitoring network sites to detect specific types of fluxes or flux patterns can be explored, as well as the impact of various sources of uncertainty on detection (e.g., Shiga et al., 2014; Fang et al., 2014; Miller et al., 2016a). Such experiments can determine how much information about the true flux field is provided by an observational network. The uncertainty reduction from the prior to the posterior estimates (see Sections 3.2 and 3.3.2) provides an overall metric for evaluating the

information provided by hypothetical observations (e.g., Chevallier et al., 2010; Baker et al., 2010; Hungershoefer et al., 2010).

Finally, through sensitivity tests, OSSEs can help to determine optimal model resolution and observational averaging for obtaining the most accurate posterior fluxes. This has been done for model temporal resolution and observational temporal

averaging (e.g., Gourdji et al., 2010). OSSEs can also be used to test the performance of the optimization of multiscale grids, which can decrease computational costs relative to regularly spaced grids (e.g., Wu et al., 2011).

## 4 Evaluation of existing diagnostics

We have presented diagnostics as an approach to the needs of quality control and of quality assurance for atmospheric inversion systems. The diagnostics that were presented in Section 3, in many ways, address this question well. The diversity of diagnostics may even give the impression that they can compensate for the lack of direct independent validation measurements described in Section 2, and thereby ensure statistical optimality of inverse modelling systems. Indeed, even uncertain parameters (hyperparameters) of the prior and observation error covariance matrices are optimisable from the assimilated data (e.g., Section 3.3.3). In most cases, however, such an interpretation would be overly optimistic. The diagnostic approaches described in Section 3 provide a crucial toolbox for evaluating and improving flux estimates obtained through the solution of atmospheric inverse problems. Without diagnostics, it is impossible to assess whether flux estimates are reliable, or to make sense of differences among alternative sets of estimates. At the same time, however, none of the presented approaches overcome the fundamental challenges described in Section 2. As such, the information provided by diagnostic tests must itself be taken with a proverbial "grain of salt," and it is equally important to be aware of the aspects of an inversion that cannot be evaluated using existing diagnostics as it is to assess those that can.

The key information lies in available measurements: diagnostics can only help to reformulate this information by bringing to light the impact of specific assumptions, in the same way that the atmospheric inversion reformulates observed concentrations in terms of surface fluxes, or that a retrieval scheme for an Earth observing system reformulates the measured radiance information into a geophysical quantity. For instance, the principle of objectively tuning error statistics for atmospheric inversions (e.g., Michalak et al., 2004; 2005) ultimately relies on disentangling deviations between prior flux assumptions and observations into components attributable to prior uncertainty versus model-data-mismatch errors. The attribution to these two components of error is based on leveraging differences in their space-time structure, however, and is made easier when the two sources of error have features that are statistically distinct (e.g., Desroziers et al., 2005). Alternatively, some of the statistics may be well known from some other information source and can then play the role of a fixed point to deduce the other ones (e.g., Kuppel et al., 2013). It is important to remember, however, that diagnostics cannot bring original information to the problem, but rather provide a framework for interpreting available information. This is particularly obvious when no real measurements are assimilated (the synthetic data experiments of Section 3.4).

The interpretation of diagnostics is also complicated by the fact that many of them are not independent of the underlying assumptions of the inversion systems themselves (e.g., independence of prior errors from model-data mismatch errors, uncorrelated nature of model-data-mismatch errors, linear observation operator, Gaussian error statistics, etc.). As a result, they may simply express the inadequacy of these assumptions rather than the misspecification of some particular component of the inversion setup. A common example is the inflation of observation error variances to compensate for neglecting

observation error correlations, which yields a too small model-data-mismatch (see Section 3.2.2) that cannot be adequately resolved without removing the decorrelation hypothesis (e.g., Chevallier, 2007).

The comparison of inversion results with independent (un-assimilated) concentration measurements (Section 3.1.1) is also partly ambiguous, because an unknown fraction of the misfit is simply caused by the chemistry and transport model that simulates the independent measurements. Similarly, the interpretation of differences between inversion results and flux estimates from bottom-up inventories (Section 3.1.2) may revolve around estimating the uncertainty of the latter (see, e.g., the diverging conclusions of Chevallier et al. (2014) and Reuter et al. (2014) about the quality of the inferred carbon sink of Europe).

Sensitivity tests about some components of the inversion systems, like the chemistry and transport model (see Section 3.3.1), are implemented in an attempt to sample the same error statistics as those specified by the model-data-mismatch and prior error covariance matrices. In practice, however, they may instead reflect different opinions about the error statistics. For instance, intercomparisons of inversion results like those of Transcom (e.g., Gurney et al., 2002, Peylin et al., 2013) form ¨ensembles of convenience¨ rather than statistically-coherent ensembles. They may underestimate the quality of state-of-the-art inversions (because some systems would underperform due to particularly coarse horizontal resolution or due to an out-dated transport simulation configuration) as well as overestimate it (because the few participants cannot sample the whole uncertainty space). To represent inversion uncertainty, inversion intercomparisons should explore the space of uncertainty widely (e.g., the ensemble would not be limited to one particular source of information for its prior fluxes for a given source-sink process) and in a balanced way (e.g., the ensemble would not oversample marginally-different versions of a single transport model at the expense of other transport model types). However, this goal is usually hampered by limited resources that favour existing set-ups over the design of systematic explorations of other plausible and defensible set-ups.

Overall then, satisfying the diagnostics described in Section 3 is, strictly speaking, neither a sufficient nor a necessary condition for optimality (see also the discussion in Talagrand 2014). The degree of usefulness of diagnostics is proportional to the amount of information that is input to them; conversely, lack of independent information can lead to problems of equifinality, where similar apparent skill is achieving through widely different setups and assumptions. In some cases, the process of identifying and improving weak components of an inverse system itself represents an inference problem that may be ill-posed or under-determined. As a result, the interpretation of diagnostics itself often requires subjective expert knowledge.

Despite their ambiguity, however, the role and diversity of diagnostics has increased over the years, and this is an important and positive development. Indeed, the diagnostics described in Section 3 have proven their practical usefulness in understanding the behaviour of inversion systems, by providing a fresh perspective on inversion results. Moreover, they can reveal, or at least suggest, the presence of hidden flaws in inversion systems by shedding light on the symptoms of these flaws. As such, they form a critical basis for the credibility of the inversion approach to flux estimation. While existing diagnostics tools have limitations, some of which are unavoidable given the challenges described in Section 2, a careful

review of the literature makes it clear that the implementation of diagnostics is a necessary step in the "exploration" of an inversion system.

## 5 Looking ahead

Atmospheric inversions are increasingly expected to contribute to national reporting of greenhouse gas emissions under future international treaties (see the discussions in Ogle et al., (2015) for biogenic emissions, Miller and Michalak (2017) for anthropogenic emissions, and Wu et al., (2016) for urban emissions). The routine run of atmospheric inversion systems will necessitate reinforcing the robustness and the transparency of their process through commonly agreed upon quality insurance and quality control procedures. In practice, this implies systematically providing reliable associated uncertainty statistics together with the posterior fluxes, and some evidence of the statistical consistency of these fluxes with the inversion assumptions. Such norms will have to rely on the systematic implementation of diagnostics of the type discussed here to a large extent, even for emerging applications like the quantification of urban emissions (McKain et al., 2012).

As we have seen in Section 4, many more measurements are needed to decrease diagnostics ambiguities. This requirement primarily relates to concentration measurements rather than flux measurements because scale mismatches usually hamper the comparison of inversions with the latter (see Section 2). A step in data density may be achieved by hypothetical low cost sensors (Wu et al., 2016) or from future satellite imagers (e.g., Rayner et al., 2014), provided these new data do not suffer from significant systematic errors. Efforts to substantially increase observational coverage are already under way (see, e.g., Climate-KIC (2017), Ciais et al., (2015)), but the feasibility of sufficiently limiting systematic errors remains to be demonstrated.

Interestingly, a (large) increase in the horizontal resolution of the inversion systems would also make it possible to incorporate direct flux measurements in the diagnostics, even when the targeted scales are coarser (see discussion in Section 2 and Lauvaux et al. (2009) or Meesters et al. (2012)). Inversion systems could also be run at very high resolution for the express purpose of comparing estimates to flux measurements. The validation with accurate flux measurements would avoid some of the ambiguity imposed by the chemistry and transport models on the concentration-based diagnostics.

This would also open up new directions for diagnostics development. For example, direct comparison to flux observations would make it possible to better assess posterior uncertainties, for instance by building on diagnostics developed in the context of ensemble prediction systems – diagnostics that have not yet been used for atmospheric inversions (e.g., the reliability diagram of Talagrand et al., 1999). These ideas were explored, for example, by Broquet et al. (2013), using aggregates of flux measurements. Among other benefits, the direct validation of the posterior uncertainties would reveal possible departures from normality for flux errors, which may be especially important in the case of systematically positive emissions (e.g., Koohkan et al., 2013). Such diagnostics would certainly help to guide future developments of inversion systems.

Taken together, it is clear that the importance of developing and implementing carefully-designed diagnostics for atmospheric inversions of long-lived greenhouse gases is only going to grow over time.

**Acknowledgments**

We acknowledge the support from the International Space Science Institute (ISSI). This publication is an outcome of the
ISSI's Working Group on "Carbon Cycle Data Assimilation: How to consistently assimilate multiple data streams". Support for Nina Randazzo was provided by the National Science Foundation under Grant No. 1342076.

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
