# Peer review of "Diagnostic methods for atmospheric inversions of long-lived greenhouse gases"

_Atmospheric Chemistry and Physics, 2016_

## Referee Comment (RC1) · Anonymous Referee #1 · 2 Nov 2016

The manuscript provides an interesting overview of the existing diagnostics to evaluate atmospheric inversions of long-lived tracers. The paper doesn't introduce any novelty in the field, but rather, it establishes a list of the existing tools. It is well written, and there is no obvious "wrong" point to comment on. I was quite pleased with Sections 1 and 2, which are a nice introduction to the topic, for non-specialists. I was unfortunately less convinced by Sections 3 and 4: although they are well written as well, I wonder what kind of reader would actually learn from it. Inverse modeling specialists are already familiar with the concepts that are presented; Non-specialists will get an idea of the diagnostics tools available, but since the paper often doesn't go much beyond listing them, they will have to read the (many) references to actually understand them.

As an example, in Section 3.1.1 (the first in which some diagnostic tools are actually presented and discussed), in 19 lines, the authors talk about: evaluation inversions

against observations left out of the inversions; evaluation inversions against observations from aircraft profiles (and as a one-line example, against vertical concentration profiles); evaluation of satellite observations constrained inversions using in-situ measurements; evaluation of in-situ observations constrained inversions using satellite measurements; evaluation against "all types of independent atmospheric observations". Each of these in less than three lines.

This is not useful to the experienced inverse modelers who are already very familiar with all this. This is not very interesting for newcomers to inverse modeling (it can be summarized in one sentence: "evaluate your results against independent data", the rest is case-specific). Finally, for specialists from other disciplines who would like to get a glimpse at how inverse models are evaluated, it quickly gets boring. Meanwhile, there are important questions that could be discussed here, but that are, in the best case, left to Section 4: comparing observations with their model counterpart is not always trivial (case of satellite observations which may require an important work of data selection, bias correction, and the application of an averaging kernel to the model fields), not always wise (comparing low-resolution model $CO_2$ fields with $CO_2$ observations in an urban environment is not so smart), and not always that useful (the implications of a bias vs. independent observations in the upper stratosphere are not the same than that of a bias in the continental boundary layer). On the other hand, not doing it is sometimes catastrophic (incorrect interpretation of inversions constrained by biased satellite data).

Some subsections of Section 3 are better, but overall, the paper would read much nicer with less references, less examples, but more detailed ones (given the pedigree of the authors, I am certain that they can easily find some from their own work, and illustrate them with a few figures). Once again, the key is to define the target readers, and what they should retain: Non specialists don't need to know of tens of examples (they won't remember them all anyway), but they need to understand correctly and completely those that are presented. Specialists might be interested in the many references, but

most of them could be moved out of the main text, perhaps to one or several tables (perhaps one for each Section 3.1, 3.2 and 3.3), as it is often done in literature reviews.

Before final publication in ACP, I would therefore recommend that the authors consider revising Section 3 and 4, keeping in mind that readers should be able to learn from it without having to read the references and/or the other papers from the special issue.

---

## Referee Comment (RC2) · Anonymous Referee #2 · 15 Nov 2016

Michalak and colleagues review the recent literature on methods to assess robustness and accuracy of atmospheric inversions of long-lived GHGs. Given the importance of inversions in present biogeochemistry and potentially in future GHG emission reduction verification, such diagnostic methods are of great relevance. After an excellent introduction on the need for diagnostics and the involved challenges, the paper reviews diagnostics applied in the literature. The diagnostics are put into context in a discussion section. I recommend to publish this work, subject to some comments given below.

When reading through the list of diagnostics, a question that repeatedly came up to me was "How well an inversion actually has to meet these diagnostics to be good enough?" For example, in Sect 3.1.1, how to translate the fit to independent data into a judgement of quality? I realised that it would be asking much to comprehensively answer this question here, and Sect 4 does discuss the limitations of the set of diagnostics.

[Figure]

Neverthess, I was wondering whether it would be helpful to put more on that already along the way, to make the paper more practical.

I feel it should be mentioned early on that the cited literature can only provide examples, because I'm sure that for most (if not all) diagnostics there are further papers which have also made good use of them, and which in some cases may even deserve credit for actually having introduced them. In this context, the restriction to papers from between 2010 and 2015 does not seem entirely appropriate to me.

I missed explicit mentioning of the "reduction of uncertainty" (1-sigma(Post)/sigma(Pri)), a diagnostic which has been being widely used by many studies, mostly in OSSEs as an alternative to the synthetic inversions explained in Sect 3.4. (In this context, it would be good to mention that the choice of foci and examples is partially subjective according to the working fields of the authors.)

Specific comments:

p 6 l 15-19: Mention already here that the robustness of column data is not yet fully established (as said later in 3.3.2), to avoid an inappropriate message.

p 6 l 30: Add "global *decadal* atm. growth rates" because this statement is not valid at yearly or shorter time scales any more.

p 7 l 1-4: The cited study is for N2O - would this also work for CO2 with both sources and sinks? (By the way, I would find it useful to mention which trace gas is being looked at in the individual examples.)

p 7 l 5-7: I find that comparisons "across inversions" are misplaced in this paragraph on comparison to "independent estimates", as inversion-inversion comparisons only allow fundamentally weaker conclusions.

p 7 l 10: The term "assessment" is so general that it remains unclear what to take from this sentence.

p 8 l 4-6: This is a complicated and unspecific formulation. What about something like "...check whether the flux adjustment by the inversion are still within the specified a-priori probability distribution".

p 8 l 9-10: Posterior concentration uncertainties can indeed be calculated in theory, but in most larger applications, this is computationally very involved in practice. I feel this should be noted.

p 9 l 20+: This has already been said in Sect 3.1.1

p 9 l 31-32: The sentence "The differences ... data." seems to be incomplete.

p 9 l 33: It remains completely unclear what "quantified via ... signal" means.

p 10 l 11-18: This paragraph unspecifically uses the term "sensitivity tests", but I assume it actually refers to synthetic-data tests. It therefore seems to better fit into Sect. 3.4.

p 10 l 31: add "regional inversions", as this is only relevant there.

p 11 l 7-11: This seems to have been said already in the previous paragraph.

p 11 l 12: add "or data set" after "of a model", as it is not always models that are being used.

p 14 l 10-11: The sentence "The ambiguity ... to them" may tentatively be true but due to its awkward formulation it remains unclear what it actually means.

p 14 l 29-31: Add e.g. ", used in conjunction with high-precision data". I disagree with the notion that low-quality data will ever be sufficient on their own, even if much larger in number.

p 15 l 8: Be specific which diagnostics this sentence is referring to, because otherwise one cannot take any information from this sentence.

Minor comments:

p 4 l 14: I find the specification "aimed at ...and patterns" obvious and thus dispensible

p 5 l 25: I find that "high level groupings of" is unnessecarily confusing and should be deleted.

p 9 l 3-4: replace "an inversion" by "the transport model"

p 11 l 26: Remove "However" as this sentence is not in opposition to the previous sentences.

p 11 l 30: Rather say "can also be used".

Typos:

p 3 l 32: "atmosphere"

p 7 l 1: "inform"

p 8 l 26: delete "a comparison of"

p 15 l 13-14: Exchange "artmospheric" and "for"

---

## Referee Comment (RC3) · Anonymous Referee #3 · 21 Nov 2016

Review of the paper entitled "Diagnostic methods for atmospheric inversions of long-lived greenhouse gases" by Michalak et al.

General comment: The paper attempts to address a challenging topic related to the validation of atmospheric inversions of greenhouse gases. In response to the increasing demand for more robust atmospheric validation tools, the authors review the existing solutions to this problem, using independent data for an indirect validation or using sensitivity experiments with different statistical metrics. The review of methods and the analysis of previous studies is quite extensive and provides a valuable overview of the current state of the art for diagnostic methods. The later section aims at evaluating these diagnostics and discusses the usefulness of these approaches related to the problems they try to address. This part of the paper suggests that most of these metrics remain insufficient to evaluate the potential problems affecting inverse

flux estimates. The authors fall short of providing suggestions trying to address these limitations, for example by recommending new measurements or methodologies to diagnose and identify them. The two main solutions proposed here are an increase in atmospheric data availability and the increase in spatial resolution to overcome representation errors when evaluating against direct flux measurements. Considering that both options are unlikely to happen in many vast areas across the world, other options should be considered to help the inverse modellers provide more robust results. I would invite the authors to 1. propose clear directions for inverse modellers to address these issues, including methodologies and strategies for measurement campaigns, and 2. suggest new/other statistical metrics to better evaluate inverse results and therefore overcome the limitations of the current metrics in inversion studies. Overall, this paper is a worthwhile contribution reviewing the current diagnostics for inversions but would need to develop this last section to provide more insights to the inversion community. Therefore, I recommend this paper for publication after addressing this problem and the following specifics comments.

Page 2 - L11: The studies cited here describe component-level surveys of equipment which are isolated in time. The term "monitored" does not reflect the lack of temporal coverage from these methods.

Section 3.3.1: Past studies (e.g. COBRA campaign, or CERES) and more recent ones (e.g. based on EnKF approaches) have tried to use meteorological and GHG data to improve or characterize transport models at continental and regional scales. Similarly, global scale models have also been compared to vertical profiles. The current section is short and would need a more complete list of studies related to transport model evaluation.

Section 3.4: A description of the most important metrics used with OSSE's would help the readers to understand the possible information that can be recovered from pseudo-data experiments. Past studies have also confused the meaning (or interpretation) of these metrics. For example, error reduction analysis may be the most useful metric

one could possibly study, but often suffers from over-confidence. Discussions may be useful in this regard, and link to the "grain of salt" compared to a proper evaluation of inversions.

Page 13 - L6-8: Should the readers conclude that these metrics are not addressing the problem? Could the authors provide more insights to explain why these metrics are insufficient? I think most inverse modelers would agree with the statement but examples of shortcomings or reasons for this failure are needed here.

Page 13 - L15-21: Do we need specific data to implement these methods? The spatial and temporal structures of errors are critical to inversions but the authors should provide more suggestions to address the separation of contributions from prior and transport model errors. This problem is non-trivial and has been studied in other fields in a more systematic fashion. Maybe references from non-GHG assimilation studies may help here.

Page 14 - L3-8: Few studies have tried to address this problem, for example the Global Carbon Project with a more coherent framework to compare inversion results. More generally, the authors could describe how to construct ensembles able to represent inversion errors. Again, possible examples from other communities (e.g. weather prediction systems) may help to find solutions, or at least, avenues that the inversion modelers could take to generate better probabilistic ensembles.

---

## Author Comment (AC2) · 21 Jan 2017

Response to Referee #2

We thank the Referee for their constructive input. We have structured our response using the following sequence, per instructions: (1) comments from Referee, (2) author response, (3) changes in manuscript.

COMMENT FROM REFEREE: Michalak and colleagues review the recent literature on methods to assess robustness and accuracy of atmospheric inversions of long-lived GHGs. Given the importance of inversions in present biogeochemistry and potentially in future GHG emission reduction verification, such diagnostic methods are of great relevance. After an excellent introduction on the need for diagnostics and the involved challenges, the paper reviews diagnostics applied in the literature. The diagnostics are

put into context in a discussion section. I recommend to publish this work, subject to some comments given below.

AUTHOR RESPONSE: We thank the Referee for their positive assessment of this work.

CHANGES IN MANUSCRIPT: None.

COMMENT FROM REFEREE: When reading through the list of diagnostics, a question that repeatedly came up to me was "How well an inversion actually has to meet these diagnostics to be good enough?" For example, in Sect 3.1.1, how to translate the fit to independent data into a judgement of quality? I realised that it would be asking much to comprehensively answer this question here, and Sect 4 does discuss the limitations of the set of diagnostics. Neverthess, I was wondering whether it would be helpful to put more on that already along the way, to make the paper more practical.

AUTHOR RESPONSE: The Referee's point / question is well taken. The question of the extent to which a given inversion has to satisfy a given metric is application-dependent, and in some cases subjective and perhaps even controversial. This adds to the complexity of applying diagnostics in this particular field.

CHANGES IN MANUSCRIPT: In revising the manuscript, we will add an overview paragraph to each subsection in Section 3 (3.1, 3.2, 3.3, 3.4) providing a clearer context and synthesis across the approaches to be presented in each subsection, and take that opportunity to touch on the question of "how good is good enough" brought up by the referee.

COMMENT FROM REFEREE: I feel it should be mentioned early on that the cited literature can only provide examples, because I'm sure that for most (if not all) diagnostics there are further papers which have also made good use of them, and which in some cases may even deserve credit for actually having introduced them. In this context, the restriction to papers from between 2010 and 2015 does not seem entirely appropriate

to me.

AUTHOR RESPONSE: The referee is correct that the manuscript cannot provide a completely comprehensive survey of existing literature. This is always a delicate and subjective balance. For example, Referee #1 actually recommended that we go in the opposite direction, significantly cutting the number of examples presented.

CHANGES IN MANUSCRIPT: In revising the manuscript, we will explicitly state that the referenced manuscripts do not represent a comprehensive set. We will also better articulate our reasoning for the selected level of detail, as also outlined in our response to Referee #1. In terms of the limitation to 2010-2015 and the decision to cite a recent paper vs. an original paper, our goal was primarily to showcase recent applications of specific types of diagnostics, rather than to present a historical view of when specific diagnostic approaches were originally proposed. We do believe that a balance needs to be struck, so in revising the manuscript we will also cite original papers where this would be beneficial, and at a minimum make sure that we do not imply that a recent paper is the original presentation of a given approach when we are in fact simply using it as an example of a contemporary application thereof. Finally, we will augment the existing list of references with some key papers from 2016.

COMMENT FROM REFEREE: I missed explicit mentioning of the "reduction of uncertainty" (1- sigma(Post)/sigma(Pri)), a diagnostic which has been being widely used by many studies, mostly in OSSEs as an alternative to the synthetic inversions explained in Sect 3.4. (In this context, it would be good to mention that the choice of foci and examples is partially subjective according to the working fields of the authors.)

AUTHOR RESPONSE: We agree that the reduction of uncertainty is frequently used in OSSEs and inversion studies. However, this metric is primarily used to assess the information content of a particular set of observations, rather than to assess the validity, self-consistency, or robustness of the inversion system itself. We did, however, briefly discuss this approach in the original Section 3.3.2.

CHANGES IN MANUSCRIPT: We will make the focus of the presented metrics clearer in the first paragraph of Section 3 and subsection 3.4.

COMMENT FROM REFEREE: Specific comments: p 6 l 15-19: Mention already here that the robustness of column data is not yet fully established (as said later in 3.3.2), to avoid an inappropriate message.

AUTHOR RESPONSE / CHANGES IN MANUSCRIPT: Agreed. We will do so.

COMMENT FROM REFEREE: p 6 l 30: Add "global *decadal* atm. growth rates" because this statement is not valid at yearly or shorter time scales any more.

AUTHOR RESPONSE / CHANGES IN MANUSCRIPT: Agreed. We will do so.

COMMENT FROM REFEREE: p 7 l 1-4: The cited study is for N2O - would this also work for CO2 with both sources and sinks? (By the way, I would find it useful to mention which trace gas is being looked at in the individual examples.)

AUTHOR RESPONSE / CHANGES IN MANUSCRIPT: Good point. We will edit to make it clear that this statement is less valid for gases such as CO2. We will also revise throughout to make target gases clearer.

COMMENT FROM REFEREE: p 7 l 5-7: I find that comparisons "across inversions" are misplaced in this paragraph on comparison to "independent estimates", as inversion-inversion comparisons only allow fundamentally weaker conclusions.

AUTHOR RESPONSE / CHANGES IN MANUSCRIPT: Agreed. We will make the change.

COMMENT FROM REFEREE: p 7 l 10: The term "assessment" is so general that it remains unclear what to take from this sentence

AUTHOR RESPONSE / CHANGES IN MANUSCRIPT: We agree that the statement was vague. The cited paper describes the comparison of the seasonal cycle of estimated CH4 mixing ratios (from an inversion constrained by in situ measurements)

to that of independent TCCON CH4 columns (both averaged over multiple TCCON locations). The large-scale agreement in this cycle is thought to support the TM5 tropopause height seasonality, because this dynamic contributes to the seasonality of column CH4. This comparison was also made for posterior CH4 columns from an inversion constrained by satellite XCH4 to determine appropriate seasonal bias correction (as explained in the next sentence of the review paper), and the agreement in the seasonal cycles between the observations and the in situ-constrained inversion posterior provides evidence that the phase shift in the satellite-constrained inversion posterior seasonal cycle is not due to a misrepresentation of tropopause height or another large scale seasonal meteorological variable in the transport model. We will clarify this in the revised manuscript.

COMMENT FROM REFEREE: p 8 l 4-6: This is a complicated and unspecific formulation. What about something like "...check whether the flux adjustment by the inversion are still within the specified a-priori probability distribution".

AUTHOR RESPONSE / CHANGES IN MANUSCRIPT: We agree that the statement was vague. A completely objective criterion is difficult to define, however. The example text provided by the referee, for example, would not work, because if one assumes a Gaussian distribution then any value is technically "within" the distribution. We will add a brief discussion of chi-squared statistics etc., but also make it clear that these metrics carry with them assumptions of their own.

COMMENT FROM REFEREE: p 8 l 9-10: Posterior concentration uncertainties can indeed be calculated in theory, but in most larger applications, this is computationally very involved in practice. I feel this should be noted.

AUTHOR RESPONSE / CHANGES IN MANUSCRIPT: Agreed. We will note this in the revision.

COMMENT FROM REFEREE: p 9 l 20+: This has already been said in Sect 3.1.1

AUTHOR RESPONSE / CHANGES IN MANUSCRIPT: The distinction between this portion of Section 3.3.2 and Section 3.1.1 is whether the additional observations are used to evaluate a posteriori fluxes (3.1.1) vs. whether the inversion is conducted multiple times, each time using a different set of observations (3.3.2). We will make this distinction clearer in the revision, and also avoid any redundant discussion.

COMMENT FROM REFEREE: p 9 l 31-32: The sentence "The differences ... data." seems to be incomplete.

AUTHOR RESPONSE / CHANGES IN MANUSCRIPT: We do not believe so. Subject: "The differences in the geographical flux patterns." Verb: "can be attributed." How: "through the use . . .."

COMMENT FROM REFEREE: p 9 l 33: It remains completely unclear what "quantified via ... signal" means.

AUTHOR RESPONSE / CHANGES IN MANUSCRIPT: We agree this was unclear. We mean that calculating the effective number of degrees of freedom provided by a given set of observations gives insight into the information content of those data with respect to fluxes. One can then use this metric to compare different (sub)sets of observations. We will make this clearer in the revision.

COMMENT FROM REFEREE: p 10 l 11-18: This paragraph unspecifically uses the term "sensitivity tests", but I assume it actually refers to synthetic-data tests. It therefore seems to better fit into Sect. 3.4.

AUTHOR RESPONSE / CHANGES IN MANUSCRIPT: We disagree. We are referring to the fact that one can run multiple "real data" inversions, each time using a different subset of available observations. We will make this clearer in the revision.

COMMENT FROM REFEREE: p 10 l 31: add "regional inversions", as this is only relevant there.

AUTHOR RESPONSE / CHANGES IN MANUSCRIPT: Agreed. We will make the

change.

COMMENT FROM REFEREE: p 11 l 7-11: This seems to have been said already in the previous paragraph.

AUTHOR RESPONSE / CHANGES IN MANUSCRIPT: Agreed. We will merge the two paragraphs and shorten the discussion where possible.

COMMENT FROM REFEREE: p 11 l 12: add "or data set" after "of a model", as it is not always models that are being used.

AUTHOR RESPONSE / CHANGES IN MANUSCRIPT: Agreed. We will make the change.

COMMENT FROM REFEREE: p 14 l 10-11: The sentence "The ambiguity ... to them" may tentatively be true but due to its awkward formulation it remains unclear what it actually means.

AUTHOR RESPONSE / CHANGES IN MANUSCRIPT: We will replace "ambiguity" by "equifinality", which better describes what we mean (the same value for a given metric can be obtained by several inversion configurations).

COMMENT FROM REFEREE: p 14 l 29-31: Add e.g. ", used in conjunction with high-precision data". I disagree with the notion that low-quality data will ever be sufficient on their own, even if much larger in number.

AUTHOR RESPONSE / CHANGES IN MANUSCRIPT: We thank the reviewer for pointing this out. We fully agree and will make the change.

COMMENT FROM REFEREE: p 15 l 8: Be specific which diagnostics this sentence is referring to, because otherwise one cannot take any information from this sentence.

AUTHOR RESPONSE / CHANGES IN MANUSCRIPT: We agree that the sentence was too vague. We will replace "(e.g., Candille and Talagrand, 2005)" by "(e.g., the reliability diagram of Talagrand et al., 1999)."

Reference: Talagrand, O., Vautard, R. and Strauss, B. (1999), Evaluation of probabilistic prediction systems. in Proceeding of workshop on predictability, p. 1-25, October 1997. European Centre for Medium-Range Weather Forecasts, Shinfield Park, Reading, Berkshire RG2 9AX, UK, http://www.ecmwf.int/sites/default/files/elibrary/1997/12555-evaluation-probabilistic-prediction-systems.pdf

COMMENTS FROM REFEREE: Minor comments:

p 4 l 14: I find the specification "aimed at ...and patterns" obvious and thus dispensible

p 5 l 25: I find that "high level groupings of" is unnessecarily confusing and should be deleted.

p 9 l 3-4: replace "an inversion" by "the transport model"

p 11 l 26: Remove "However" as this sentence is not in opposition to the previous sentences.

p 11 l 30: Rather say "can also be used".

AUTHOR RESPONSE / CHANGES IN MANUSCRIPT: We agree with all the minor comments above, and will make edits accordingly.

COMMENTS FROM REFEREE: Typos:

p 3 l 32: "atmosphere"

p 7 l 1: "inform"

p 8 l 26: delete "a comparison of"

p 15 l 13-14: Exchange "artmospheric" and "for"

AUTHOR RESPONSE / CHANGES IN MANUSCRIPT: We agree with all the minor comments above, and will make edits accordingly.

---

## Author Comment (AC3) · 21 Jan 2017

Response to Referee #3

We thank the Referee for their constructive input. We have structured our response using the following sequence, per instructions: (1) comments from Referee, (2) author response, (3) changes in manuscript.

COMMENT FROM REFEREE: General comment: The paper attempts to address a challenging topic related to the validation of atmospheric inversions of greenhouse gases. In response to the increasing demand for more robust atmospheric validation tools, the authors review the existing solutions to this problem, using independent data for an indirect validation or using sensitivity experiments with different statistical metrics. The review of methods and the analysis of previous studies is quite extensive

and provides a valuable overview of the current state of the art for diagnostic methods. The later section aims at evaluating these diagnostics and discusses the usefulness of these approaches related to the problems they try to address. This part of the paper suggests that most of these metrics remain insufficient to evaluate the potential problems affecting inverse flux estimates. The authors fall short of providing suggestions trying to address these limitations, for example by recommending new measurements or methodologies to diagnose and identify them. The two main solutions proposed here are an increase in atmospheric data availability and the increase in spatial resolution to overcome representation errors when evaluating against direct flux measurements. Considering that both options are unlikely to happen in many vast areas across the world, other options should be considered to help the inverse modellers provide more robust results. I would invite the authors to 1. propose clear directions for inverse modellers to address these issues, including methodologies and strategies for measurement campaigns, and 2. suggest new/other statistical metrics to better evaluate inverse results and therefore overcome the limitations of the current metrics in inversion studies. Overall, this paper is a worthwhile contribution reviewing the current diagnostics for inversions but would need to develop this last section to provide more insights to the inversion community. Therefore, I recommend this paper for publication after addressing this problem and the following specifics comments

AUTHOR RESPONSE: Following other authors (e.g., Desroziers et al. 2005 and Talagrand 2014) quoted in our text, we made it clear in Section 4 that diagnostics are ambiguous in a way that is inversely proportional to the amount of information which is input to them. In the revised version, we will refer to the principle of equifinality. The current challenge therefore does not lie in the definition of new metrics, but rather in the increase of the amount of information. We therefore propose (i) to increase the horizontal resolution in order to exploit some of the existing data that can hardly be used now, and (ii) to increase the measurement density. Item (i) is already happening (see, e.g., the comparison between atmospheric inversions and tower/aircraft flux measurements in Lauvaux et al. 2009, Meesters et al. 2012, Broquet et al. 2013),

while item (ii) is being actively prepared by some companies and by space agencies (e.g., http://www.copernicus.eu/sites/default/files/library/CO2_Report_22Oct2015.pdf). We therefore argue that these prospects are both necessary and achievable in the near future. In the meantime, we also argue that there is much inspiration and much quality to be gained in using the existing data better in most inversion systems by following some of the diagnostics that are presented in our text (we used the expression "crucial tool box" twice).

CHANGES IN MANUSCRIPT: We will develop Section 5 in order to illustrate the fact that the way forward is achievable and already happening.

COMMENT FROM REFEREE: Page 2 - L11: The studies cited here describe component-level surveys of equipment which are isolated in time. The term "monitored" does not reflect the lack of temporal coverage from these methods.

AUTHOR RESPONSE: Agreed.

CHANGES IN MANUSCRIPT: We will mention that providing temporal coverage is an additional challenge.

COMMENT FROM REFEREE: Section 3.3.1: Past studies (e.g. COBRA campaign, or CERES) and more recent ones (e.g. based on EnKF approaches) have tried to use meteorological and GHG data to improve or characterize transport models at continental and regional scales. Similarly, global scale models have also been compared to vertical profiles. The current section is short and would need a more complete list of studies related to transport model evaluation.

AUTHOR RESPONSE: The referee's point is well taken. We had intended for this section to describe the use of unused atmospheric observations for diagnosing inversion systems as a whole, but we agree that the evaluation of atmospheric transport models is a crucial component thereof.

CHANGES IN MANUSCRIPT: We will add a paragraph focusing specifically on the use

of atmospheric observations for evaluating atmospheric transport models.

COMMENT FROM REFEREE: Section 3.4: A description of the most important metrics used with OSSE's would help the readers to understand the possible information that can be recovered from pseudodata experiments. Past studies have also confused the meaning (or interpretation) of these metrics. For example, error reduction analysis may be the most useful metric one could possibly study, but often suffers from over-confidence. Discussions may be useful in this regard, and link to the "grain of salt" compared to a proper evaluation of inversions.

AUTHOR RESPONSE: Agreed. With regard to the error reduction metric in particular, we agree that the reduction of uncertainty is frequently used in OSSEs. However, this metric is primarily used to assess the information content of a particular set of observations, rather than to assess the validity, self-consistency, or robustness of the inversion system itself. This is the reason for which it was not discussed in subsection 3.4.

CHANGES IN MANUSCRIPT: We will add a description of metrics used as part of OSSEs for diagnosing inversions systems.

COMMENT FROM REFEREE: Page 13 - L6-8: Should the readers conclude that these metrics are not addressing the problem? Could the authors provide more insights to explain why these metrics are insufficient? I think most inverse modelers would agree with the statement but examples of shortcomings or reasons for this failure are needed here.

AUTHOR RESPONSE: In our presentation, we have introduced diagnostics as an an-swer to the needs of "quality control (. . .) (i.e., the evaluation of the flux estimates) [and of] (. . .) quality assurance (i.e., the evaluation of the estimation process that yielded the flux estimates)" for the inversion systems. In this sense, diagnostics address the problem well. If we ask them for more, such as identifying what is going wrong in a system for which some diagnostics show a warning, we may be limited by the amount

of information actually available within or outside the inverse system under study. This was discussed, in part, in Section 2, which presented some of the unique challenges of developing, applying, and interpreting diagnostics for atmospheric inverse problems.

CHANGES IN MANUSCRIPT: We will recall the purpose of diagnostics at the start of Section 4. We will also formulate the limitations of diagnostics in terms of their capability to infer some property of an inversion system, based on the well- or ill-posedness of that particular inference problem.

COMMENT FROM REFEREE: Page 13 - L15-21: Do we need specific data to implement these methods? The spatial and temporal structures of errors are critical to inversions but the authors should provide more suggestions to address the separation of contributions from prior and transport model errors. This problem is non-trivial and has been studied in other fields in a more systematic fashion. Maybe references from non-GHG assimilation studies may help here.

AUTHOR RESPONSE: To be implemented, these methods do not need other data than the assimilated ones. They have been used in many non GHG assimilation studies, and the example that we give in page 13 L. 17 (Desroziers et al. 2005) actually comes from numerical weather prediction. Recent application examples in this field could be given (e.g., Stewart et al. 2014), but may not help much here. In any case, their capability always depends on the information available from the data. The referee's point about diagnosing the validity of assumptions about prior error vs. transport model errors is well taken, but is already address to the extent possible in Section 3.

Reference: Stewart, L. M., Dance, S. L., Nichols, N. K., Eyre, J. R. and Cameron, J. (2014), Estimating interchannel observation-error correlations for IASI radiance data in the Met Office system†. Q.J.R. Meteorol. Soc., 140: 1236–1244. doi:10.1002/qj.2211

CHANGES IN MANUSCRIPT: None.

COMMENT FROM REFEREE: Page 14 - L3-8: Few studies have tried to address this

problem, for example the Global Carbon Project with a more coherent framework to compare inversion results. More generally, the authors could describe how to construct ensembles able to represent inversion errors. Again, possible examples from other communities (e.g. weather prediction systems) may help to find solutions, or at least, avenues that the inversion modelers could take to generate better probabilistic ensembles.

AUTHOR RESPONSE: An ensemble of inversion results represents inversion errors well provided that the corresponding ensemble of inversion set-ups explores the space of uncertainty widely (e.g., the ensemble would not be limited to one particular source of information for its prior fluxes for a given source-sink process) and in a balanced way (e.g., the ensemble would not oversample marginally-different versions of a single transport model at the expense of other transport model types). In practice, this goal is usually hampered by limited resources that favor existing set-ups over the design of systematic explorations of other plausible and defensible set-ups. These statements are general and not limited to the GHG community.

CHANGES IN MANUSCRIPT: We will expand the discussion to better capture this challenge.

---

## Author Response (AR1)

Response to Referee #1

We thank the Referee for their constructive input. We have structured our response using the following sequence, per instructions: (1) comments from Referee, (2) author response, (3) changes in manuscript.

COMMENT FROM REFEREE: The manuscript provides an interesting overview of the existing diagnostics to evaluate atmospheric inversions of long-lived tracers. The paper doesn't introduce any novelty in the field, but rather, it establishes a list of the existing tools. It is well written, and there is no obvious "wrong" point to comment on. I was quite pleased with Sections 1 and 2, which are a nice introduction to the topic, for non-specialists. I was unfortunately less convinced by Sections 3 and 4: although they are well written as well, I wonder what kind of reader would actually learn from it. Inverse modeling specialists are already familiar with the concepts that are presented; Non-specialists will get an idea of the diagnostics tools available, but since the paper often doesn't go much beyond listing them, they will have to read the (many) references to actually understand them.

AUTHOR RESPONSE: We thank the referee for recognizing the value and intent of the work. With regard to Sections 3 and 4, our goal is two-fold. Although we agree that inverse modeling specialists will already be familiar with some (or many) of the concepts presented, we doubt that any specialist would be familiar with the full spectrum of approaches presented here, unless they themselves had conducted a full review of the literature. Speaking on behalf of the three specialists who authored this manuscript, although each of us was familiar with many of the approaches we describe, we each also learned about some that were new to us. For non-specialists, we agree that they would need to read additional references in order to get a deeper understanding of a particular approach. We consider this a strength rather than a weakness of the work. In essence, in Section 3 we are providing a guided tour of the literature, which would allow a non-specialist to know exactly where to go for a more in depth presentation of any particular approach.

CHANGES IN MANUSCRIPT: In the revised manuscript, we have more clearly enunciated the dual target audience for this review in Section 1. We have also implemented changes to put the approaches within a clearer context, as described in more detail in other responses below.

COMMENT FROM REFEREE: As an example, in Section 3.1.1 (the first in which some diagnostic tools are actually presented and discussed), in 19 lines, the authors talk about: evaluation inversions against observations left out of the inversions; evaluation inversions against observations from aircraft profiles (and as a one-line example, against vertical concentration profiles); evaluation of satellite observations constrained inversions using in-situ measurements; evaluation of in-situ observations constrained inversions using satellite measurements; evaluation against "all types of independent atmospheric observations". Each of these in less than three lines.

This is not useful to the experienced inverse modelers who are already very familiar with all this. This is not very interesting for newcomers to inverse modeling (it can be summarized in one sentence: "evaluate your results against independent data", the rest is case-specific). Finally, for specialists from other disciplines who would like to get a glimpse at how inverse models are evaluated, it quickly gets boring. Meanwhile, there are important questions that could be discussed here, but that are, in the best case, left to Section 4: comparing observations with their model counterpart is not always trivial (case of satellite observations which may require an important work of data selection, bias correction, and the application of an averaging kernel to the model fields), not always wise (comparing low-resolution model CO2 fields with CO2 observations in an urban environment is not so smart), and not always that useful (the implications of a bias vs. independent observations in the upper stratosphere are not the same than that of a bias in the continental boundary layer). On the other hand, not doing it is sometimes catastrophic (incorrect interpretation of inversions constrained by biased satellite data).

AUTHOR RESPONSE: The referee's point is well taken. As described in the first response above, we believe that there is value to presenting a broad ranging set of references and variations on diagnostics approaches, for both specialists and non-specialists. This goes with our vision of this manuscript as a roadmap to the existing literature. At the same time, we agree with the referee that in some instances the desire to be thorough came at the expenses of synthesis and interpretation.

CHANGES IN MANUSCRIPT: In revising the manuscript, we have examined each sub-section in Section 3 with the goal of keeping detail to the extent it is useful, but at the same time restructuring the discussion in a way that synthesizes information more effectively, as with the example listed by the referee. The main changes come at the start of each subsection (3.1, 3.2, 3.3, 3.4), where we have presented more context for the subset of diagnostics to be presented.

COMMENT FROM REFEREE: Some subsections of Section 3 are better, but overall, the paper would read much nicer with less references, less examples, but more detailed ones (given the pedigree of the authors, I am certain that they can easily find some from their own work, and illustrate them with a few figures). Once again, the key is to define the target readers, and what they should retain: Non specialists don't need to know of tens of examples (they won't remember them all anyway), but they need to understand correctly and completely those that are presented. Specialists might be interested in the many references, but most of them could be moved out of the main text, perhaps to one or several tables (perhaps one for each Section 3.1, 3.2 and 3.3), as it is often done in literature reviews. Before final publication in ACP, I would therefore recommend that the authors consider revising Section 3 and 4, keeping in mind that readers should be able to learn from it without having to read the references and/or the other papers from the special issue.

AUTHOR RESPONSE: We agree with the referee that a review of this sort could fundamentally take one of two forms. The first is, as the referee described, an in depth look at a small selected set of prototypical examples. The second is, as we have done here, a more comprehensive overview of the literature. There are advantages and disadvantages to each. We made the choice to use the second model in part specifically to make it easy for readers to dig deeper if they chose to, by looking up the references included in the manuscript.

CHANGES IN MANUSCRIPT: With the above in mind, we have decided not to restructure the manuscript by limiting discussion to a few prototypical examples and putting the remainder of references in a table. That being said, we take the referee's concerns to heart, and have revised Section 3 to provide more context, synthesis, and interpretation, to the extent possible without substantially increasing the overall length of the manuscript. Section 4 can actually be read without knowing the ten papers referenced in it: these have been included only as examples and options for further reading; hence, these are are now all preceded by either "e.g." or "see also".

Response to Referee #2

We thank the Referee for their constructive input. We have structured our response using the following sequence, per instructions: (1) comments from Referee, (2) author response, (3) changes in manuscript.

COMMENT FROM REFEREE: Michalak and colleagues review the recent literature on methods to assess robustness and accuracy of atmospheric inversions of long-lived GHGs. Given the importance of inversions in present biogeochemistry and potentially in future GHG emission reduction verification, such diagnostic methods are of great relevance. After an excellent introduction on the need for diagnostics and the involved challenges, the paper reviews diagnostics applied in the literature. The diagnostics are put into context in a discussion section. I recommend to publish this work, subject to some comments given below.

AUTHOR RESPONSE: We thank the Referee for their positive assessment of this work.

CHANGES IN MANUSCRIPT: None.

COMMENT FROM REFEREE: When reading through the list of diagnostics, a question that repeatedly came up to me was "How well an inversion actually has to meet these diagnostics to be good enough?" For example, in Sect 3.1.1, how to translate the fit to independent data into a judgement of quality? I realised that it would be asking much to comprehensively answer this question here, and Sect 4 does discuss the limitations of the set of diagnostics. Nevertheless, I was wondering whether it would be helpful to put more on that already along the way, to make the paper more practical.

AUTHOR RESPONSE: The Referee's point / question is well taken. The question of the extent to which a given inversion has to satisfy a given metric is application-dependent, and in some cases subjective and perhaps even controversial. This adds to the complexity of applying diagnostics in this particular field.

CHANGES IN MANUSCRIPT: In revising the manuscript, we have added an overview paragraph to each subsection in Section 3 (3.1, 3.2, 3.3, 3.4) providing a clearer context and synthesis across the approaches to be presented in each subsection, and take that opportunity to touch on the question of "how good is good enough" brought up by the referee.

COMMENT FROM REFEREE: I feel it should be mentioned early on that the cited literature can only provide examples, because I'm sure that for most (if not all) diagnostics there are further papers which have also made good use of them, and which in some cases may even deserve credit for actually having introduced them. In this context, the restriction to papers from between 2010 and 2015 does not seem entirely appropriate to me.

AUTHOR RESPONSE: The referee is correct that the manuscript cannot provide a completely comprehensive survey of existing literature. This is always a delicate and subjective balance. For example, Referee #1 actually recommended that we go in the opposite direction, significantly cutting the number of examples presented.

CHANGES IN MANUSCRIPT: In revising the manuscript, we have explicitly stated that the referenced manuscripts do not represent a comprehensive set (see revised Section 3.0). We have also better articulated our reasoning for the selected level of detail, as also outlined in our response to Referee #1 (see revised Section 1). In terms of the limitation to 2010-2015 and the decision to cite a recent paper vs. an original paper, our goal was primarily to showcase recent applications of specific types of diagnostics, rather than to present a historical view of when specific diagnostic approaches were originally proposed. We have clarified this in the revised Section 3.0. We do believe that a balance needs to be struck, so in revising the manuscript we have also cited original papers where beneficial, and at a minimum made sure that we do not imply that a recent paper is the original presentation of a given approach when we are in fact simply using it as an example of a contemporary application thereof (see revised Section 3.0 and details in subsequent subsections of Section 3). Finally, we have augmented the existing list of references with some key papers from 2016.

COMMENT FROM REFEREE: I missed explicit mentioning of the "reduction of uncertainty" (1- sigma(Post)/sigma(Pri)), a diagnostic which has been being widely used by many studies, mostly in OSSEs as an alternative to the synthetic inversions explained in Sect 3.4. (In this context, it would be good to mention that the choice of foci and examples is partially subjective according to the working fields of the authors.)

AUTHOR RESPONSE: We agree that the reduction of uncertainty is frequently used in OSSEs and inversion studies. However, this metric is primarily used to assess the information content of a particular set of observations, rather than to assess the validity, self-consistency, or robustness of the inversion system itself. We did, however, briefly discuss this approach in the original Section 3.3.2.

CHANGES IN MANUSCRIPT: We have made the focus of the presented metrics clearer in the revised Section 3.0, but also added a brief comment about the "reduction in uncertainty" metric in Section 3.4.

COMMENT FROM REFEREE: Specific comments:
p 6 l 15-19: Mention already here that the robustness of column data is not yet fully established (as said later in 3.3.2), to avoid an inappropriate message.

AUTHOR RESPONSE / CHANGES IN MANUSCRIPT: Agreed. We have done so.

COMMENT FROM REFEREE: p 6 l 30: Add "global *decadal* atm. growth rates" because this statement is not valid at yearly or shorter time scales any more.

AUTHOR RESPONSE / CHANGES IN MANUSCRIPT: Agreed.  We have done so.

COMMENT FROM REFEREE: p 7 l 1-4: The cited study is for N2O - would this also work for CO2 with both sources and sinks? (By the way, I would find it useful to mention which trace gas is being looked at in the individual examples.)

AUTHOR RESPONSE / CHANGES IN MANUSCRIPT: Good point.  We have edited to make it clear that this statement is less valid for gases such as CO2.  We have also revised throughout to make target gases clearer where appropriate.

COMMENT FROM REFEREE: p 7 l 5-7: I find that comparisons "across inversions" are misplaced in this paragraph on comparison to "independent estimates", as inversion-inversion comparisons only allow fundamentally weaker conclusions.

AUTHOR RESPONSE / CHANGES IN MANUSCRIPT: Agreed.  We have removed the reference to comparisons from one inversion to another.

COMMENT FROM REFEREE: p 7 l 10: The term "assessment" is so general that it remains unclear what to take from this sentence

AUTHOR RESPONSE / CHANGES IN MANUSCRIPT:  We agree that the statement was vague. The cited paper describes the comparison of the seasonal cycle of estimated CH4 mixing ratios (from an inversion constrained by in situ measurements) to that of independent TCCON CH4 columns (both averaged over multiple TCCON locations).  The large-scale agreement in this cycle is thought to support the TM5 tropopause height seasonality, because this dynamic contributes to the seasonality of column CH4.  This comparison was also made for posterior CH4 columns from an inversion constrained by satellite XCH4 to determine appropriate seasonal bias correction (as explained in the next sentence of the review paper), and the agreement in the seasonal cycles between the observations and the in situ-constrained inversion posterior provides evidence that the phase shift in the satellite-constrained inversion posterior seasonal cycle is not due to a misrepresentation of tropopause height or another large scale seasonal meteorological variable in the transport model.  We have clarified this in the revised manuscript.

COMMENT FROM REFEREE: p 8 l 4-6: This is a complicated and unspecific formulation. What about something like "...check whether the flux adjustment by the inversion are still within the specified a-priori probability distribution".

AUTHOR RESPONSE / CHANGES IN MANUSCRIPT: We agree that the statement was vague.  A completely objective criterion is difficult to define, however.  The example text provided by the referee, for example, would not work, because if one assumes a

Gaussian distribution then any value is technically "within" the distribution. We will add a brief discussion of chi-squared statistics etc., but also make it clear that these metrics carry with them assumptions of their own.

COMMENT FROM REFEREE: p 8 l 9-10: Posterior concentration uncertainties can indeed be calculated in theory, but in most larger applications, this is computationally very involved in practice. I feel this should be noted.

AUTHOR RESPONSE / CHANGES IN MANUSCRIPT: Agreed. We will note this in the revision.

COMMENT FROM REFEREE: p 9 l 20+: This has already been said in Sect 3.1.1

AUTHOR RESPONSE / CHANGES IN MANUSCRIPT: The distinction between this portion of Section 3.3.2 and Section 3.1.1 is whether the additional observations are used to evaluate a posteriori fluxes (3.1.1) vs. whether the inversion is conducted multiple times, each time using a different set of observations (3.3.2). We have made this distinction clearer in the revision, and also avoided any redundant discussion.

COMMENT FROM REFEREE: p 9 l 31-32: The sentence "The differences ... data." seems to be incomplete.

AUTHOR RESPONSE / CHANGES IN MANUSCRIPT: We do not believe so. Subject: "The differences in the geographical flux patterns." Verb: "can be attributed." How: "through the use ...."

COMMENT FROM REFEREE: p 9 l 33: It remains completely unclear what "quantified via ... signal" means.

AUTHOR RESPONSE / CHANGES IN MANUSCRIPT: We agree this was unclear. We mean that calculating the effective number of degrees of freedom provided by a given set of observations gives insight into the information content of those data with respect to fluxes. One can then use this metric to compare different (sub)sets of observations. We have made this clearer in the revision.

COMMENT FROM REFEREE: p 10 l 11-18: This paragraph unspecifically uses the term "sensitivity tests", but I assume it actually refers to synthetic-data tests. It therefore seems to better fit into Sect. 3.4.

AUTHOR RESPONSE / CHANGES IN MANUSCRIPT: We disagree. We are referring to the fact that one can run multiple "real data" inversions, each time using a different subset of available observations. We have made this clearer in the revision.

COMMENT FROM REFEREE: p 10 l 31: add "regional inversions", as this is only relevant there.

AUTHOR RESPONSE / CHANGES IN MANUSCRIPT:  Agreed.  We have made the change.

COMMENT FROM REFEREE: p 11 l 7-11: This seems to have been said already in the previous paragraph.

AUTHOR RESPONSE / CHANGES IN MANUSCRIPT:  Agreed.  We have deleted redundant text.

COMMENT FROM REFEREE: p 11 l 12: add "or data set" after "of a model", as it is not always models that are being used.

AUTHOR RESPONSE / CHANGES IN MANUSCRIPT:  Agreed. We have made the change.

COMMENT FROM REFEREE: p 14 l 10-11: The sentence "The ambiguity ... to them" may tentatively be true but due to its awkward formulation it remains unclear what it actually means.

AUTHOR RESPONSE / CHANGES IN MANUSCRIPT: We have replaced "ambiguity" by "equifinality", which better describes what we mean (the same value for a given metric can be obtained by several inversion configurations).

COMMENT FROM REFEREE: p 14 l 29-31: Add e.g. ", used in conjunction with high-precision data". I disagree with the notion that low-quality data will ever be sufficient on their own, even if much larger in number.

AUTHOR RESPONSE / CHANGES IN MANUSCRIPT: We thank the reviewer for pointing this out. We fully agree and have made the change.

COMMENT FROM REFEREE: p 15 l 8: Be specific which diagnostics this sentence is referring to, because otherwise one cannot take any information from this sentence.

AUTHOR RESPONSE / CHANGES IN MANUSCRIPT: We agree that the sentence was too vague. We have replaced "(e.g., Candille and Talagrand, 2005)" by "(e.g., the reliability diagram of Talagrand et al., 1999)."

. We therefore argue that these prospects are both necessary and achievable in the near future. In the meantime, we also argue that there is much inspiration and much quality to be gained in using the existing data better in most inversion systems by following some of the diagnostics that are presented in our text (we used the expression "crucial tool box" twice).

CHANGES IN MANUSCRIPT: We have developed Section 5 in order (i) to state the need of providing both uncertainty statistics with the posterior fluxes and some evidence of the statistical consistency of these fluxes with the inversion assumptions, (ii) to develop the need of increased measurement density (we have made the distinction between flux and concentration measurements and have added a brief discussion about possible systematic errors in future measurement types), (iii) to refer to Lauvaux et al. (2009) and Meesters et al. (2012) as existing examples of the possibility to compare high-resolution inversion results with flux measurements.

COMMENT FROM REFEREE:  Page 2 - L11: The studies cited here describe component-level surveys of equipment which are isolated in time. The term "monitored" does not reflect the lack of temporal coverage from these methods.

AUTHOR RESPONSE: Agreed.

CHANGES IN MANUSCRIPT: We now mention both the spatial (i.e. point) and temporal (i.e. episodic, discrete) aspects of these observations.

COMMENT FROM REFEREE:  Section 3.3.1: Past studies (e.g. COBRA campaign, or CERES) and more recent ones (e.g. based on EnKF approaches) have tried to use meteorological and GHG data to improve or characterize transport models at continental and regional scales. Sim
current section is short and would need a more complete list of studies related to transport model evaluation.

AUTHOR RESPONSE: The referee's point is well taken.  We had intended for this section to describe the use of unused atmospheric observations for diagnosing inversion systems as a whole, but we agree that the evaluation of atmospheric transport models is a crucial component thereof.

CHANGES IN MANUSCRIPT: We have added a paragraph focusing specifically on the use of atmospheric observations for evaluating atmospheric transport models in Section 3.1.1, as we felt this was a more natural location for the information presented therein.

COMMENT FROM REFEREE:  Section 3.4: A description of the most important metrics used with OSSE's would help the readers to understand the possible information that can be recovered from pseudodata experiments. Past studies have also confused the meaning (or interpretation) of these metrics. For example, error reduction analysis may be the most useful metric one could possibly study, but often suffers from over-confidence. Discussions may be useful in this regard, and link to the "grain of salt" compared to a proper evaluation of inversions.

AUTHOR RESPONSE: Agreed. With regard to the error reduction metric in particular, we agree that the reduction of uncertainty is frequently used in OSSEs. However, this metric is primarily used to assess the information content of a particular set of observations, rather than to assess the validity, self-consistency, or robustness of the inversion system itself. This is the reason for which it was not originally discussed in subsection 3.4.

CHANGES IN MANUSCRIPT: We have added a description of metrics used as part of OSSEs for diagnosing inversions systems. We also added a brief comment about overall error reduction in the revised Section 3.4.

COMMENT FROM REFEREE: Page 13 - L6-8: Should the readers conclude that these metrics are not addressing the problem? Could the authors provide more insights to explain why these metrics are insufficient? I think most inverse modelers would agree with the statement but examples of shortcomings or reasons for this failure are needed here.

AUTHOR RESPONSE: In our presentation, we have introduced diagnostics as an answer to the needs of "quality control (…) (i.e., the evaluation of the flux estimates) [and of] (…) quality assurance (i.e., the evaluation of the estimation process that yielded the flux estimates)" for the inversion systems. In this sense, diagnostics address the problem well. If we ask them for more, such as identifying what is going wrong in a system for which some diagnostics show a warning, we may be limited by the amount of information actually available within or outside the inverse system under study. This was discussed, in part, in Section 2, which presented some of the unique challenges of developing, applying, and interpreting diagnostics for atmospheric inverse problems.

CHANGES IN MANUSCRIPT: We have recalled the purpose of diagnostics at the start of Section 4. We have also formulated the limitations of diagnostics at the end of the section in terms of their capability to infer some property of an inversion system, based on the well- or ill-posedness of that particular inference problem.

COMMENT FROM REFEREE: Page 13 - L15-21: Do we need specific data to implement these methods? The spatial and temporal structures of errors are critical to inversions but the authors should provide more suggestions to address the separation of contributions from prior and transport model errors. This problem is non-trivial and has been studied in other fields in a more systematic fashion. Maybe references from non-GHG assimilation studies may help here.

AUTHOR RESPONSE: To be implemented, these methods do not need other data than the assimilated ones. They have been used in many non GHG assimilation studies, and the example that we give in page 13 L. 17 (Desroziers et al. 2005) actually comes from numerical weather prediction.  Recent application examples in this field could be given (e.g., Stewart et al. 2014), but may not help much here. In any case, their capability always depends on the information available from the data.  The referee's point about diagnosing the validity of assumptions about prior error vs. transport model errors is well taken, but is already address to the extent possible in Section 3.

Reference:
Stewart, L. M., Dance, S. L., Nichols, N. K., Eyre, J. R. and Cameron, J. (2014), Estimating interchannel observation-error correlations for IASI radiance data in the Met Office system†. Q.J.R. Meteorol. Soc., 140: 1236–1244. doi:10.1002/qj.2211

CHANGES IN MANUSCRIPT: None.

COMMENT FROM REFEREE:  Page 14 - L3-8: Few studies have tried to address this problem, for example the Global Carbon Project with a more coherent framework to compare inversion results. More generally, the authors could describe how to construct ensembles able to represent inversion errors. Again, possible examples from other communities (e.g. weather prediction systems) may help to find solutions, or at least, avenues that the inversion modelers could take to generate better probabilistic ensembles.

AUTHOR RESPONSE: An ensemble of inversion results represents inversion errors well provided that the corresponding ensemble of inversion set-ups explores the space of uncertainty widely (e.g., the ensemble would not be limited to one particular source of information for its prior fluxes for a given source-sink process) and in a balanced way (e.g., the ensemble would not oversample marginally-different versions of a single transport model at the expense of other transport model types). In practice, this goal is usually hampered by limited resources that favor existing set-ups over the design of systematic explorations of other plausible and defensible set-ups. These statements are general and not limited to the GHG community.

CHANGES IN MANUSCRIPT: We have expanded the discussion with the words of our response to the comment to better capture this challenge.

[revised manuscript text omitted]